# Effect of implementing of the IDEAL discharge model on satisfaction of patient referred to trauma emergency department

Zahra Moradi Rekabdar Kalaiee[1], Raziyeh Ghafouri [2]*, Mitra Zandi[2], Malihe Nasiri[3]

**1** Student Research Committee, School of Nursing & Midwifery, Shahid Beheshti University of Medical Sciences, Tehran, Iran, **2** Department of Medical and Surgical Nursing, School of Nursing and Midwifery, Shahid Beheshti University of Medical Sciences, Tehran, Iran, **3** Department of Basic Sciences, School of Nursing & Midwifery, Shahid Beheshti University of Medical Sciences, Tehran, Iran

\* ghafouri@sbmu.ac.ir, raziehghafouri@gmail.com

**Data Availability Statement:** All relevant data are within the manuscript and its Supporting Information files.

## Abstract

### Background

Patient education at the time of discharge is one of the most important challenges in the emergency department. This study aimed to evaluate the Effect of implementing the IDEAL, or integrated discharge model, on the satisfaction of patients referred to the trauma emergency department.

### Methods

This quasi-experimental study was conducted on the patients referred to the trauma emergency department of Imam Hossein Hospital in Tehran. Eighty-six patients were recruited from January 20, 2023, until July 30, 2023. In the intervention group, the educational content was taught using the IDEAL (Inclusion, Discussion, Education, Assessment, Listening) or integrated discharge model, and in the control group, it was taught using the emergency department routine method. The satisfaction questionnaire of the emergency department was completed before and after the education in both groups and compared between the two groups. The data were analyzed by using SPSS (version 20) software.

### Results

The results showed that out of the 86 participating patients, 52 (60.5%) were male and 34 (39.5%), with a mean (Standard Deviation) of 39.14 (10.89) years old. Demographic characteristics were homogeneous between the two groups (P > 0.05). The mean (standard deviation) of satisfaction of the participants after education, totally was 63.56 (16.21), in the intervention group it was 77.37 (7.95), and in the Control group it was 49.74 (8.84). The mean (SD) participants satisfaction on arrival at the emergency department in the intervention group was 19.16 (2.75) and in the control group was 13.51 (2.51), during hospitalization in the intervention group was 10.72 (1.77) and in the Control group 6.74 (1.81), discharge time in the intervention group 14.51 (2.93) and in the control group 2.93 (2.04), Overall satisfaction with nursing care in intervention group 13.85 (2.46) and in the control group 8.46

**Funding:** The authors received no specific funding for this work.

**Competing interests:** The authors declared no conflicts of interest with respect to the research, authorship, and/or publication of this article.

(2.41), Overall satisfaction with medical procedures in the intervention group 12.81 (2.73) and in the control group 8.58 (3.20) and Overall patient satisfaction in the intervention group 2.27 (1.81) and 41.4 (1.66) in the control group. An independent T-test was used to compare satisfaction and its dimensions in two groups, and there was a statistically significant difference between the two groups (P<0.01).

## Conclusion

The study results showed a statistically significant difference in the satisfaction in the intervention and control groups, so it can be concluded that conducting the integrated discharge model is effective in increasing the satisfaction of trauma patients. Therefore, it is recommended to use this educational method to increase patient satisfaction and decrease readmission rates.

## Introduction

The emergency room is the most important department in any hospital, and its performance has a great impact on other departments and the satisfaction of clients. It is referred to as the heart of the hospital [1–3]. Trauma is one of the four causes of death in developing countries and the second cause of death among young people in the world [4, 5].

Clients' satisfaction is the best and most important indicator to measure the quality of services provided [3] and its promotion is one of the goals of medical centers [6, 7]. Clients with higher satisfaction from the care provider are more likely to follow their medical instructions [2, 3, 7, 8].

One of the most important factors in decreasing satisfaction among emergency departments is the time of discharge [9, 10], so that people feel worried and anxious because they go to new situations and do not have enough knowledge about the disease, prognosis and necessary care related to them. Failure to provide education in these cases will reduce the satisfaction of the patient and his family [2, 10, 11].

Discharge is a period of transition from hospital to home in which responsibility for care is transferred from the care providers during hospitalization to the patient, family, and primary care physician [10, 12, 13]. Surveys have shown that discharge programs in the emergency department are not optimal [12, 14, 15]. One of the ways to improve it is to use the IDEAL, or integrated discharge model [16, 17].

The International Agency for Quality and Research in Health has presented the IDEAL discharge model [18], and discharge planning starts from the time of admission of the patient and involves anticipating the needs of the patient and his family and planning to meet these needs after discharge. The key elements of comprehensive discharge planning include the patient and family, discharge planning, education for the patient and family, evaluation of the amount of education, and effective listening [18, 19].

The involvement of the patient's family and caregivers in the training of the discharge time will lead to better follow-up of the patients, reduce their return to the emergency room of medical centers, and have fewer complications [19]. When education is provided to patients' families and caregivers, key gaps in self-care knowledge and warning signs are reduced [18, 19]. Enhancing caregiver engagement at discharge, particularly related to caregiver problem-

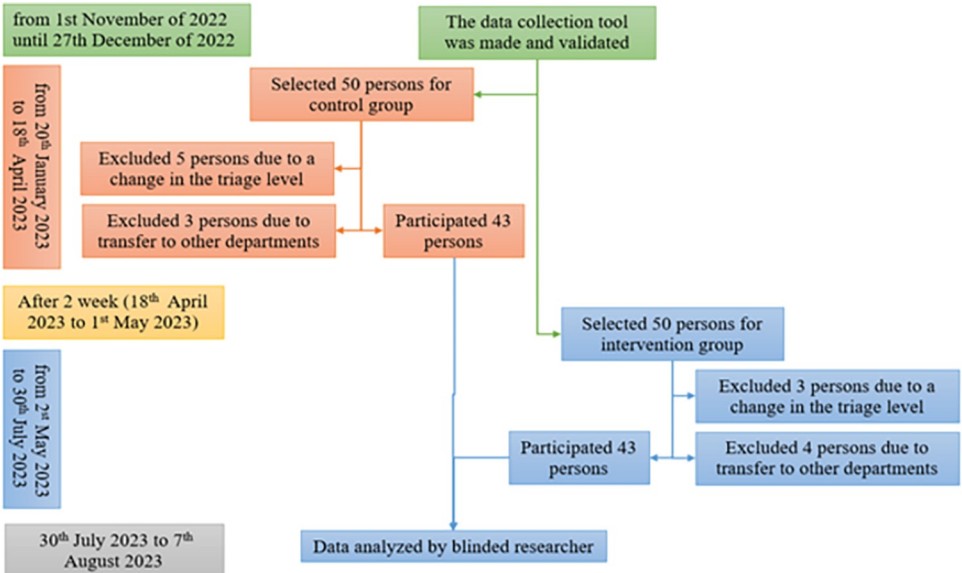

**Fig 1. The process of the study.**

solving assessment, planning, and post-discharge support, in efforts that seek to improve care transitions and post-discharge outcomes [19].

The implementation of the IDEAL (Inclusion, Discussion, Education, Assessment, Listening), or integrated discharge model, has been effective in reducing patients' anxiety [18, 19], medication compliance [16] and readmission [17]. Considering the advantages of the integrated discharge model, including educational planning from the beginning of admission with the active participation of the patient and its family-centeredness, the use of this model can be helpful in training clients in the emergency department [19], the present study was conducted to aim to investigate the impact of the implementation of the IDEAL, or integrated discharge model, on the satisfaction of patients hospitalized in the trauma emergency department of Imam Hossein Medical Education Center in Tehran.

## Methods

The present study was a quasi-experimental interventional study and was conducted as a clinical trial with blinding. The study was conducted in the emergency department of Imam Hossein Hospital in Tehran from 2022 until 2023. The data collection tool was made and validated from November 1st, 2022, until 27 December 2022, and sampling was done from January 20th, 2023, until July 30th, 2023. The process of research was demonstrated in Fig 1.

Imam Hossein Hospital is an academic and referral trauma center, with 630 approved beds, 521 active beds, 30 emergency rooms and 30 ICU beds. The number of patients hospitalized due to trauma in this hospital is 30–100 patients per day, and annually more than 30,000 patients are hospitalized due to trauma in this hospital.

## Participants

The research population included patients hospitalized due to trauma in the emergency department. Inclusion criteria were age 65–18 years, hospitalized due to trauma in the emergency department, levels 3 and 4, with reading and writing literacy, absence of pain during education, awareness of place and time with a level of consciousness of 15 with the Glasgow

Coma Scale (GCS), and a normal vital sign (systolic blood pressure of 100–120 mmHg, dia-
stolic blood pressure 70–80 mmHg, a resting heart rate of 60 to 80 beats per minute (BPM),
the respiratory rate 12 to 18 breaths per minute), No need for mechanical or artificial ventila-
tion, having no history of similar education program, having suitable physical and mental con-
ditions for education, and they were discharged from the emergency room. Exclusion criteria
included unwillingness to continue cooperation and participation in the study, a change in the
patient's condition, and changing his triage level to levels 1 and 2, referring the patient from
the emergency department to other departments.

According to the results of Stevens et al.'s study [14] and assuming equal variance in the
two groups and considering the difference value of 1.25 units (d), standard deviation (σ) 2,
and assuming statistical type 1 error (α) of 0.05 and the statistical power of the study is equal
to 0.9, and using the following formula:

$$n = \frac{\left(Z_{1-\alpha/2} + Z_{1-\beta}\right)^2 * 2 * \sigma^2}{d^2}$$

$$\alpha = 0.05 \ (z_{0.975} = 1.96),$$

$$\beta = 0.8 \ (z_{0.8} = 0.84),$$

$$d = 1.25$$

$$\sigma = 2$$

$$n = (1.96 + 0.84)2 * 2^2)/1.25^2 = 40.14$$

The sample size in each group was 40 persons; by adding about 10% lost to follow-up, the
number of samples in each group was estimated to be 43 persons.

Sampling was done using a conventional method. In order not to publish information
between the intervention and control groups, first the control group and then the intervention
group were sampled. The two groups were statistically analyzed and matched in terms of age,
gender, reason for hospitalization, triage level and time of hospitalization in the emergency
room.

During the data collection for each group, 50 eligible people were selected to participate in
the research. In the control group, 5 people were excluded from the study due to a change in
the triage level, 3 people were excluded from the study due to transfers to other departments,
and finally 43 people participated in the study. In the intervention group, 3 people were
excluded from the study due to a change in the level of triage, 4 people were excluded from the
study due to transfers to other departments, and finally 43 people participated in the study. A
total of 86 people recruited in the research.

## Data collection tools

In this research, the data collection tool includes demographic information and a question-
naire made by the researcher on the satisfaction of clients from the emergency department
with 22 items in 6 areas of satisfaction: on arrival (5 items), during hospitalization (3 items),
discharge time (4 items), overall satisfaction with nursing care (4 items), overall satisfaction
with medical procedures (4 items), and overall satisfaction (2 items). The demographic

**Table 1. Results of the quantitative content validity.**

| CVI | | | CVR |
|---|---|---|---|
| relevance | simplicity | Ambiguity | |
| 0.92 | 0.88 | 0.87 | 0.88 |

information questionnaire included the name and surname, age, sex, education, marital status and economic status of the clients.

The questionnaire items were designed based on a review of literature and the opinions of emergency room specialists and nurses. In order to determine the validity of the questionnaires used, the method of determining the face and content validity was used, and in the reliability check, the Cronbach's alpha method was used [18, 19]. In order to check the face validity and qualitative content validity, the tools were given to 10 faculty members of the Faculty of Nursing and Midwifery of Shahid Beheshti University of Medical Sciences and specialists and nurses with experience working in the emergency department, and after collecting their opinions, 22 items in 6 dimensions were included in the questionnaire.

The content validity was done using a quantitative method by using the opinions of 10 members of the Nursing and Midwifery School of Shahid Beheshti University of Medical Sciences and experts and nurses with experience working in the emergency room. The simplicity, ambiguity, relevance of the items, content validity index (CVI), and necessity of the items were calculated by the content validity ratio (CVR). The content validity index is higher than 0.7 in each case; simplicity, ambiguity, and relevance in each item are acceptable. The content validity ratio based on the number of participating professors (at least 10 people) is 0.49, which is the minimum acceptable according to the Lauwshe table [20, 21]. The formula for the content validity ratio (CVR) is $CVR = (Ne - N/2)/(N/2)$. Table 1 shows the result of the quantitative content validity.

Cronbach's alpha coefficient was used for the reliability of the tool. Cronbach's alpha coefficient above 0.7 is good, 0.3–0.7 is good, and less than 0.3 is poor [20, 21]. The overall Cronbach's alpha of the tool was 0.72, which indicated the appropriate reliability of the research tool.

## Data collection

After selecting the participants, the objectives of the research were first explained to them, and after obtaining the approval of the Ethics and Registration Committee at the Clinical Research Center of Iran, sampling was done from January 20, 2023, until July 30, 2023. In order to prevent the dissemination of information between the intervention and control groups, first the control group and then the intervention group were sampled. The data of the control group was collected from January 2023 until April 18, 2023; the data of the intervention group was collected after 2 weeks, from May 2023, to July 30th, 2023.

Clients were assured about the importance of conducting research and the confidentiality of information; informed consent was obtained, and then their demographic information was completed. Clients and their caregivers in the intervention group, in addition to receiving educational materials according to the routine of the department, were trained using the consolidated discharge method, and the clients of the control group received the necessary training according to the routine of the department and were discharged from the hospital. The satisfaction questionnaires of clients in two intervention and control groups were completed before the education and after education, at the time of discharge.

The sample was selected by RG, education was presented by ZM, data was collected by MZ and data was analyzed by MN. The statistical analyst and data collector researcher was not aware of the allocation of intervention and control groups.

## Intervention

In order to provide education based on the IDEAL (Inclusion, Discussion, Education, Assessment, Listening) discharge model, education was provided to clients and their caregivers. Patient education in intervention was presented before discharge in a 1-hour meeting with clients and their caregivers. The educational content is based on the self-care nursing model and using the existing guidelines and scientific principles of patient education with the aim of empowering clients to take care of themselves in the field of drugs (reason for use, precautions related to drugs, prescription instructions, guidance In case of forgetting to take, drug use during pregnancy, complications and conditions of preparation and storage) and general care (control of vital signs, diagnosis of infection symptoms, diet recommendations, referral routine, lifestyle modification, measures to prevent complications, how to use from health care services and compliance with physical activity), (Education) by face-to-face method and with discussion (Discussion), according to their needs, understanding and disease situation (Assessment) with the aim of empowering clients in self-care to clients and their caregivers (Inclusion) was presented, At the end of 10 minutes, the questions of the clients and their caregivers were answered (Listening).

## Data analysis

The collected data were statistically analyzed by Statistical Package for Social Sciences version 20 (SPSS 20). In order to compare the mean of quantitative variables in the intervention and control groups, the independent two-sample t test was used. Also, the non-parametric Mann-Whitney U test was used to compare the mean of the variables of duration of illness and duration of hospitalization in the intervention and control groups. In order to compare the average satisfaction score and its ranges before and after the educational intervention, in each of the studied groups, a paired T-test was used. In order to compare the frequency distribution of qualitative variables in the intervention and control groups, a chi-square test or Fisher's exact test was used.

The normality of the frequency distribution of quantitative variables was evaluated with the Kolmogorov-Smirnov non-parametric test, and the equality of variance of the groups was also evaluated with Levene's test. The significance level in the tests was considered to be 0.05.

## Ethical considerations

The present study has been registered with the code of ethics code IR.SBMU.PHARMACY. REC.1400.334 by the research ethics committee of the Faculty of Nursing, Midwifery and Pharmacy of Shahid Beheshti University of Medical Sciences and in the International Clinical Trial Registration Center of Iran under the number IRCT20210131050189N3.

The authors guarantee that they have followed the ethical principles stated in the Declaration of Helsinki (to protect the life, health, dignity, integrity, right to self-determination, privacy, and confidentiality of personal information of research subjects) in all stages of the research. To observe the ethical considerations, hospitals agreed. To observe the ethical considerations, the research goals and procedures were elucidated to the participants, they were assured of information anonymity and confidentiality, and informed written consent was obtained from each participant. They participated in the study voluntarily and could leave the study at any stage.

## Results

The participants were 86 patients participating in the study, with an average age of 39.14 ± 10.89 years. The demographic characteristics of the participants (gender, age, marital status, level of education, triage level, and time spent in the emergency room) had the same distribution in the two research groups, and there was no statistically significant difference between the two groups (P > 0.05). The average (Standard Deviation) age of the research participants in the intervention group was 39.65 (11.78) years, and in the control group it was 38.62 (10.02) years. According to the U-Man-Whitney test, there was no statistically significant difference between the two groups of participants in the study in the intervention and control groups in terms of average age (P > 0.05). The average (Standard Deviation) hospitalization time in the emergency room was 8.12 (4.61) hours in the intervention group and 10.09 (7.29) hours in the control group. According to the U-Man-Whitney test, there was no statistically significant difference between the two groups of participants in the study, in the intervention and control groups, in terms of hospitalization time in the emergency room (P > 0.05). Table 2 shows the demographic characteristics of the participants.

The mean (standard deviation) of before-education satisfaction of the emergency department among the participants was 43.65 (12.65); in the intervention group, it was 42.48 (6.44); and in the control group, it was 44.81 (16.72). An independent T-test was used to compare before-education satisfaction in two groups, and there was not a statistically significant difference between the two groups (P = 0.39, t = 0.85). Also, there was not a statistically significant difference between the satisfaction dimensions of the two groups (P > 0.05) (Table 3).

The mean (standard deviation) of after-education satisfaction of the emergency department among the participants was 63.56 (16.21), in the intervention group it was 77.37 (7.95), and in

**Table 2. Demographic characteristics of the participants.**

| Demographic characteristic | | Control | | Intervention | | Test Results |
|---|---|---|---|---|---|---|
| | | % | N | % | N | |
| Gender | Male | 53.5 | 20 | 23.6 | 13 | $\chi^2 = 1.75$ df = 1 P = 0.27 |
| | Female | 46.5 | 23 | 67.4 | 29 | |
| | Total | 100 | 43 | 100 | 43 | |
| Job | Employee | 23.3 | 10 | 14 | 6 | $\chi^2 = 4.02$ df = 3 P = 0.25 |
| | Merchant job | 46.5 | 20 | 39.5 | 17 | |
| | Housekeeper | 18.6 | 8 | 32.2 | 16 | |
| | Retired | 11.6 | 5 | 9.3 | 4 | |
| | Total | 100 | 43 | 100 | 43 | |
| Education | Elementary school | 7 | 3 | 9.3 | 4 | $\chi^2 = 7.31$ df = 4 P = 0.12 |
| | High school | 16.3 | 7 | 9.3 | 4 | |
| | Associate Degree | 23.3 | 10 | 11.6 | 5 | |
| | Bachelor's degree | 39.5 | 17 | 65.1 | 28 | |
| | Graduate | 14 | 6 | 4.7 | 2 | |
| | Total | 100 | 43 | 100 | 43 | |
| The economic situation (monthly income) | Sufficient | 14 | 6 | 18.6 | 8 | $\chi^2 = 1.24$ df = 2 P = 0.58 |
| | Moderate | 60.5 | 26 | 65.1 | 28 | |
| | Low | 25.6 | 11 | 16.3 | 7 | |
| | Total | 100 | 43 | 100 | 43 | |
| Triage level | 3 | 55.8 | 24 | 52.2 | 22 | $\chi^2 = 0.18$ df = 1 P = 0.82 |
| | 4 | 44.2 | 19 | 48.8 | 21 | |
| | Total | 100 | 43 | 100 | 43 | |

**Table 3. Comparison of the before-education satisfaction of the emergency department in the intervention and control groups.**

| Satisfaction items | | Control | Intervention | Test Results |
|---|---|---|---|---|
| On arrival | Mean | 11.11 | 9.63 | t = -1.25 |
| | SD | 7.27 | 2.85 | P = 0.21 |
| During hospitalization | Mean | 5.81 | 5.34 | t = -0.89 |
| | SD | 2.92 | 1.75 | P = 0.37 |
| Discharge time | Mean | 7.46 | 7.81 | t = 0.58 |
| | SD | 3.40 | 1.95 | P = 0.56 |
| Overall satisfaction with nursing care | Mean | 8.18 | 8.46 | t = 0.41 |
| | SD | 3.84 | 2.27 | P = 0.68 |
| Overall satisfaction with medical procedures | Mean | 8.16 | 7.41 | t = 0.99 |
| | SD | 4.45 | 2.09 | P = 0.49 |
| Overall patient satisfaction | Mean | 4.06 | 3.81 | t = 0.68 |
| | SD | 2.02 | 1.36 | P = 0.001 |

the control group it was 49.74 (8.84). An independent T-test was used to compare after-education satisfaction in two groups. The results of the independent t-test indicated a significant difference between the after-education satisfaction of the intervention and control groups (P = 0.001; t = 15.23). Also, there was a statistically significant difference between the dimensions of satisfaction of the two groups (P<0.01) (Table 4).

The results of the paired t-test indicated a significant difference in the satisfaction of the clients before and after education in the intervention group (P < 0.01), while no significant difference was seen in the control group (P > 0.05).

## Discussion

This study was conducted with the aim of determining the effect of the implementation of the consolidated discharge model on the satisfaction of the patients who went to the trauma emergency department of the Imam Hossein Medical Education Center in Tehran, and the research findings based on the results of the independent t-test indicate a statistically significant difference in satisfaction at the time of discharge in both the intervention and control groups (P<0.01), so the results shows the effectiveness of implementing the IDEAL discharge model in increasing the satisfaction of trauma clients.

**Table 4. Comparison of after-education satisfaction of the emergency department in the intervention and control groups.**

| Satisfaction items | | Control | Intervention | Test Results |
|---|---|---|---|---|
| On arrival | Mean | 13.51 | 19.16 | t = 9.95 |
| | SD | 2.51 | 2.75 | P = 0.001 |
| During hospitalization | Mean | 6.74 | 10.72 | t = 8.46 |
| | SD | 1.81 | 1.77 | P = 0.001 |
| Discharge time | Mean | 2.93 | 14.51 | t = 11.89 |
| | SD | 2/04 | 2.93 | P = 0.001 |
| Overall satisfaction with nursing care | Mean | 8.46 | 13.85 | t = 9.73 |
| | SD | 2.41 | 2.46 | P = 0.001 |
| Overall satisfaction with medical procedures | Mean | 8.58 | 12.81 | t = 6.59 |
| | SD | 3.20 | 2.73 | P = 0.001 |
| Overall patient satisfaction | Mean | 4.41 | 7.27 | t = 7.61 |
| | SD | 1.66 | 1.81 | P = 0.001 |

The combined discharge model was investigated in a quasi-experimental study with the aim of investigating the effect of applying the IDEAL discharge model on the level of medication compliance of clients with congestive heart failure hospitalized in the cardiac care unit (CCU) by Rezaeifar et al. [22]. The researchers in that study have also stated that the implementation of the IDEAL discharge model has been able to significantly decrease complications of medical procedures, which is consistent with our study and indicates the effectiveness of the IDEAL discharge model for discharge [22]. Also, Baghaei et al. mentioned the effectiveness of the IDEAL discharge model and stated that its implementation was able to effectively reduce the anxiety of heart attack patients [23].

The results of Waniga et al. study were in line with our study, which, concluded that the implementation of the IDEAL discharge instructions by the doctor and nurse could improve the patient's empowerment of the discharge process [24]. Comin-Colet et al. stated that the main components of the IDEAL model include initial intervention in the hospitalization stage, planned discharge and regular planning for continuous follow-up of patients [25]. The results of their studies are consistent with the results of the present study.

The effect of other educational interventions related to patient discharge on patient satisfaction in the emergency department has been investigated in various studies, which mostly indicate the positive effect of these interventions on increasing patient satisfaction [12–14]. Stevens and colleagues, in their study entitled "Fast track patients' satisfaction, compliance and confidence with emergency department discharge planning," stated that patient education based on their needs from hospitalization to discharge is effective in increasing their satisfaction, trust and reducing complications [14], as well as reducing their readmission [14, 17]. Shakeri et al. in their study stated that the implementation of educational intervention for patient self-care empowerment has a significant role in increasing the level of patient satisfaction [26].

The reason for the increase in patient satisfaction in the interventional group of this study, may be was related to the fact that the training was adjusted according to the clinical condition of the patients and based on their needs, and most of the patients were given the opportunity to participate in the discussion, as well as the training in a dynamic manner. It has been done and has high flexibility. Therefore, it can be concluded that the IDEAL discharge model by the nurse is a suitable solution to improve patients' satisfaction, and patient education based on the IDEAL discharge model can empower these patients in their self-care, be effective by itself, and, as a result, increase their satisfaction. Therefore, it is recommended to use this educational method for better and more efficient training of patients and, as a result, to increase their satisfaction. One of the strengths of the IDEAL discharge model is enhancing caregiver engagement at discharge, particularly related to caregiver problem-solving assessment, planning, and post-discharge support. In efforts that seek to improve care transitions and post-discharge outcomes, we recommend that they evaluate their engagements in education.

## Conclusion

The results of the research indicated a statistically significant difference in the satisfaction of the discharge time in the two intervention and control groups, so it can be concluded that training patients using the IDEAL discharge model training method by nurses is a suitable solution to improve patient satisfaction, as well as that training Based on the IDEAL discharge model, it can be effective in empowering these patients in self-care and thus increasing their satisfaction. Therefore, it is recommended to use this educational method for better and more efficient training of patients and, as a result, to increase their satisfaction and decrease readmissions.

### Research limitations

One of the limitations of the study was the spread of COVID-19 disease and the increase in hospitalization of COVID-19 patients in specialized departments, which led to an increase in sampling time. There were no other uncontrollable limitations that could interfere with the research process.

One of the strengths of the study was conducting the research in an academic center and referring trauma patients with numerous admissions. In the present study, outpatients were not included in the study. It is suggested to examine the IDEAL discharge model in outpatients as well; it seems that it is effective for outpatients who visit the emergency room and reduces their confusion about self-care. Outpatients also need a comprehensive discharge model due to the limited visit time.

## Supporting information

**S1 Checklist. CONSORT 2010 checklist of information to include when reporting a randomised trial\*.**
(DOC)

**S1 Data.**
(SAV)

**S1 File.**
(DOCX)

**S2 File.**
(PDF)

**S3 File.**
(DOC)

## Acknowledgments

We appreciate and thank all the participants and staff of the emergency department of the Imam Hossein Medical Educational Center in Tehran.

## Author Contributions

**Conceptualization:** Zahra Moradi Rekabdar Kalaiee, Raziyeh Ghafouri.

**Data curation:** Zahra Moradi Rekabdar Kalaiee, Raziyeh Ghafouri, Mitra Zandi, Malihe Nasiri.

**Formal analysis:** Malihe Nasiri.

**Investigation:** Zahra Moradi Rekabdar Kalaiee.

**Methodology:** Raziyeh Ghafouri, Mitra Zandi, Malihe Nasiri.

**Project administration:** Raziyeh Ghafouri, Mitra Zandi.

**Supervision:** Raziyeh Ghafouri.

**Validation:** Raziyeh Ghafouri.

**Visualization:** Zahra Moradi Rekabdar Kalaiee.

**Writing – original draft:** Zahra Moradi Rekabdar Kalaiee, Raziyeh Ghafouri.

**Writing – review & editing:** Zahra Moradi Rekabdar Kalaiee, Raziyeh Ghafouri, Mitra Zandi, Malihe Nasiri.

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
