## [Decision Letter · Decision Letter 0]

31 Jan 2024

PONE-D-23-29600Effect of Implementing of the Integrate Discharge Model on Satisfaction of Patient Referred to Trauma Emergency DepartmentPLOS ONE

Dear Dr. Ghafouri,

Thank you for submitting your manuscript to PLOS ONE. After careful consideration, we feel that it has merit but does not fully meet PLOS ONE’s publication criteria as it currently stands. Therefore, we invite you to submit a revised version of the manuscript that addresses the points raised during the review process.

We look forward to receiving your revised manuscript.

Kind regards,

Milad Khorasani, PhD

Academic Editor

PLOS ONE

Journal Requirements:

2. Thank you for stating the following financial disclosure: "No"

3. Thank you for stating the following in your Competing Interests section:  "NO authors have competing interests".

4. We note that your Data Availability Statement is currently as follows: "All relevant data are within the manuscript and its Supporting Information files."

5. We are unable to open your Supporting Information file "Data.sav". Please kindly revise as necessary and re-upload.

Reviewers' comments:

Reviewer's Responses to Questions

**Comments to the Author**

1. Is the manuscript technically sound, and do the data support the conclusions?

Reviewer #1: Yes

Reviewer #2: No

Reviewer #3: Partly

2. Has the statistical analysis been performed appropriately and rigorously? 

Reviewer #1: Yes

Reviewer #2: No

Reviewer #3: Yes

3. Have the authors made all data underlying the findings in their manuscript fully available?

Reviewer #1: Yes

Reviewer #2: No

Reviewer #3: Yes

4. Is the manuscript presented in an intelligible fashion and written in standard English?

Reviewer #1: Yes

Reviewer #2: No

Reviewer #3: No

5. Review Comments to the Author

Reviewer #1: 1- Please compare your survey to similar surveys in the literature.

2- Please mention the weak and strong points of your study

3- I recommend the authors discuss following article in the discussion:

I- Sarbazi E, Sadeghi-Bazargani H, Farahbakhsh M, Ala A, Soleimanpour H. Psychometric properties of trust in trauma care in an emergency department tool. Eur J Trauma Emerg Surg. 2023 Aug 21. doi: 10.1007/s00068-023-02348-z.

II- Rahmani F, Rezazadeh F, Ala A, Soleimanpour M, Mehdizadeh R, Soleimanpour H. Evaluation of Overcrowding of Emergency Department in Imam Reza Hospital in 2015 by Implementing 2 Scales: NEDOCS and EDWIN. Iran Red Crescent Med J. 2017 June; 19(6):e15609.

III- Soleimanpour H, Gholipouri C, Salarilak S, Raoufi P, Vahidi RG, Rouhi AJ, Ghafouri RR, Soleimanpour M. Emergency department patient satisfaction survey in Imam Reza Hospital, Tabriz, Iran. Int J Emerg Med. 2011 Jan 27;4:2. doi: 10.1186/1865-1380-1-2.

Reviewer #2: I reviewed the article entitled “Effect of Implementing of the Integrate Discharge Model on Satisfaction of Patient Referred to Trauma Emergency Department” by Kalaiee et al. submitted to PLOS ONE (Manuscript Number: PONE-D-23-29600). In this observational study involving 86 injured patients, the authors mainly investigated that the effect of implementing of the integrate discharge model on patients' satisfaction. They found that implementing of the integrate discharge model was associated with the increased patients' satisfaction, measured by self-reported questionnaire. From these observations, they claimed the usefulness of integrated discharge model in patients with trauma.

First, the reviewer pays respect for the Authors' tremendous effort spent on this manuscript. However, there are numerous concerns with the data presentation, design as well as the methodology. My concerns are listed below:

1

Much of the abstract is currently spent on background and methods. Please provide in the abstract an informative and balanced summary of what was done and what was found. At the current form, description regarding results is poor, and intervention used in this study is not clear enough.

2

This observational study does not follow STROBE Statement Guidelines (https://www.equator-network.org/reporting-guidelines/strobe/). The authors should respect the basic rule of scientific writing.

3

Please indicate the study’s design with a commonly used term in the title, according to the STROBE check list. The title also should be more specific. One example is therefore “Effect of Implementing of the Integrate Discharge Model on Satisfaction of Patient Referred to Trauma Emergency Department: a retrospective observational study at a single medical institution in Iran"

Introduction

4

At the introduction section, the authors should clarify how the information collected can be used to solve the current problems for trauma.

5

At the end of the introduction section, please state the any prespecified hypotheses according to the STROBE check list.

Methods

6

Please describe settings and locations more in details (e.g. a tertiary hospital, academic hospital, referral trauma center, number of hospital and ICU beds, and number of annual trauma admission, etc.) where the data were collected. This information should help readers to depict the context of this study more accurately. This reviewer thinks annual trauma volume is especially important because it can greatly affect the quality of trauma care [1, 2], and may be confounding factors in the analyses.

7

The Ethical approval section should include the relevant date of the approval.

8

The authors should describe the intervention used in this study greater in details. As the current form, it is difficult to replicate your study even by other skilled researchers.

9

Please give the characteristic of the data source. How did you assure the quality of data? Since this is an observational study, quality assurance is of vitally important.

10

Study period was just half ot the year (January to July, 2023. Why did you employ such short duration?

11

Clearly define all exposures, predictors, potential confounders, and effect modifiers.

12

Describe all statistical methods, including those used to control for confounding.

13

Describe any efforts to address potential sources of bias, according to the STROBE checklist. For example, blinding is one of the attractive methods to reduce above mentioned biased assessment. If done, please provide who was blinded and how.

14

Who planned this study, who collected data, and who conducted the statistical analysis? I think if the same researchers are involved in study planning, data collecting, outcome measurement, and statistical analysis, there is a theoretical risk of biased assessment.

15

The authors did not provide the information how the questionnaire used here was developed. How did you assure the scientific validity of this survey? As is the current form, the reviewer and readers of this journal cannot judge whether the questionnaire used here is validated or not. This reviewer thinks planning phase is the most important process of the survey that requires detailed description. How did you select the items of your questionnaires? Did you referred to relevant studies conducted in other countries or other setting when developing the questionnaire? In addition, the authors should provide the rationale of outcome measurements.

16

Please provide the rationale of the items included in this survey more in details.

17

Please provide an English version of the full questionnaires used in this study to facilitate readers' understanding regarding this project.

18

The authors should describe relevant dates, including periods of questionnaires development, questionnaires distribution, follow up, and collection.

19

Please explain how missing data were addressed. Did authors use complete dataset?

20

A flow diagram reporting the numbers of individuals at each stage of study—eg numbers potentially eligible, is missing.

22

Table 1. Baseline characteristics of the participants

Many vital information is missing. Give characteristics of study participants such as vital sings including GCS score, blood pressure, and respiratory rate, need of endotracheal intubation and mechanical ventilation, Charlson Comorbidity Index, trauma etiology (penetrating or blunt), Abbreviated Injury Scale of each body parts, TRISS based probability of survival etc. At this current form, many readers including myself find it difficult to image the characteristics of study subjects. In addition, these variables would have confounded the results. There are too many unmeasured confounders. This is the serious flaw of your manuscript. The authors should adjust for such important confounders to provide more reliable data.

Discussion

23

Much of the discussion section is simply the list of previous studies, restate or rephrase the results and background that have already described in results and introduction section. Most parts of discussion are too speculative, and not based on the data obtained in this study. This reviewer thinks the discussion is not thought evoking one. In addition, the discussion section should indicate how the findings of this study can be used to solve the current problems. How and for what do we use these results presented here to improve the current trauma care and why?

24

The limitation section needs substantial revision. Please discuss limitations of the study, taking into account sources of potential bias or imprecision. Consider the important limitations and do not just list them but consider their relevance and how they might bias the results. Discuss both direction and magnitude of any potential bias.

25

What is the strength of this observational study? Please indicate after the limitation section.

26

Please discuss the generalizability (external validity) and potential clinical implications for practice of the study results.

27

Please indicate future research direction more in details, immediately after limitation section.

Conclusion

28

The conclusion section is just rephrased of the observed findings. The study implications are not shown in the conclusion section.

How to use this conclusion to improve the clinical outcome of the patients? Please indicate.

Minor points

29

Keep abbreviations to a minimum. Do not use non-standard abbreviations unless they appear at least three times in the text.

30

The authors should provide the minimal anonymized data set used in this manuscript, according to the journal's policy.

Although the number of criticisms listed above, this reviewer should however state that it is laudable that this work is derived from huge efforts made by the authors, who are working as the frontline healthcare professionals. The reviewer respects the authors’ time and effort spent on this manuscript, and the authors ‘patience and professionalism in dealing with my comments.

References

1. MacKenzie EJ, Rivara FP, Jurkovich GJ, Nathens AB, Frey KP, Egleston BL, Salkever DS, Scharfstein DO.A national evaluation of the effect of trauma-center care on mortality.N Engl J Med. 2006,26;354:366-78.

2. Minei JP, Fabian TC, Guffey DM, Newgard CD, Bulger EM, Brasel KJ, Sperry JL, MacDonald RD. Increased trauma center volume is associated with improved survival after severe injury: results of a Resuscitation Outcomes Consortium study. Ann Surg. 2014, 260:456-64.

Reviewer #3: The main problem of this manuscript is the ambiguity in the sampling method, the method of randomly assignment of samples to control and test groups. When and by whom were the samples selected? The formula for calculating the sample size is not clear and numbers have not been placed in it? The numerical value of alpha and sigma is not mentioned.

Considering the nature of the emergency department and the short stay of patients, has it been possible to train them? What was the role of caregivers in this study and why was it not included in the statistical analysis? Also, this manuscript needs basic revision in terms of written English language. In table number 1, the sum of control and test samples is not correct in some parts. Finally, many of the references are old, and if possible, newer references should be used instead them. Reference number 16 does not have the year of publication. The reference writing format style needs to be edited.

6. PLOS authors have the option to publish the peer review history of their article (what does this mean?). If published, this will include your full peer review and any attached files.

Reviewer #1: No

Reviewer #2: No

Reviewer #3: **Yes: **Esmail Khodadadi

---

## [Author Response · Author response to Decision Letter 0]

1 Mar 2024

Dear editor of journal PlOS ONE 

thanks a lot for your valuable comments. we tried considered comments and improved the manuscript.

---

## [Decision Letter · Decision Letter 1]

1 Apr 2024

PONE-D-23-29600R1Effect of Implementing of the IDEAL Discharge Model on Satisfaction of Patient Referred to Trauma Emergency DepartmentPLOS ONE

Dear Dr. Ghafouri,

Thank you for submitting your manuscript to PLOS ONE. After careful consideration, we feel that it has merit but does not fully meet PLOS ONE’s publication criteria as it currently stands. Therefore, we invite you to submit a revised version of the manuscript that addresses the points raised during the review process.

We look forward to receiving your revised manuscript.

Kind regards,

Milad Khorasani, PhD

Academic Editor

PLOS ONE

Reviewers' comments:

Reviewer's Responses to Questions

**Comments to the Author**

1. If the authors have adequately addressed your comments raised in a previous round of review and you feel that this manuscript is now acceptable for publication, you may indicate that here to bypass the “Comments to the Author” section, enter your conflict of interest statement in the “Confidential to Editor” section, and submit your "Accept" recommendation.

Reviewer #1: All comments have been addressed

Reviewer #3: All comments have been addressed

Reviewer #4: (No Response)

2. Is the manuscript technically sound, and do the data support the conclusions?

Reviewer #1: Yes

Reviewer #3: Yes

Reviewer #4: No

3. Has the statistical analysis been performed appropriately and rigorously? 

Reviewer #1: Yes

Reviewer #3: Yes

Reviewer #4: No

4. Have the authors made all data underlying the findings in their manuscript fully available?

Reviewer #1: Yes

Reviewer #3: Yes

Reviewer #4: Yes

5. Is the manuscript presented in an intelligible fashion and written in standard English?

Reviewer #1: Yes

Reviewer #3: Yes

Reviewer #4: No

6. Review Comments to the Author

Reviewer #1: The respected authors revised the article in the best manners. The article is suitable for publication.

Reviewer #3: Congratulations on your perfect work. The reviewer's comments revision seems to have been well done. Your study is beneficial for Health professionals in Emergency Department in Hospitals.

Reviewer #4: Abstract: Further review by an English native speaker is needed. e.g. "Eighty-size patients were participated" should be Eighty-six patients participated. and this sentence should mention the patients were recruited between these dates, rather than all patients involved the whole period.

Some methods are specified in the results section, include independent t-test in the methods (but shouldn't it be a paired t-test as each patient has a pre and post value? Please specify the statistical methods used and confirm they were appropriate.

Methods pg 5: why was it a quasi-experimental interventional study? Isn't it an interventional study?

line 111: 3,0000 is not the standard way a number is presented: do you mean 30,000?

Power calculation: what is the difference of 10 between. Clarify if it in the mean overall score between groups at the discharge visit?

you state in the text d=10 unites, but then on line 133 it is 1.25. Please clarify.

line 135 has too many equals signs, and how does 40/14 = 40?

line 140: matched: this means different things statistically and in a sampling manner. How did it work - the matching - was it used to recruit particular participants in the second group? Was it a coincidence that both groups went from 50 to 43? i.e. did recruitment stop when the 43rd patient completed study assessments?

pg 11 - the first paragraph seems to be explaining the acronym IDEAL - the EAL is clear, but what about the I and D?

line 233 and 237 and 241: should this be p>0.05?

units for time in emergency room?

table 3: is satisfaction of the emergency department made up of the sum of the other 6 or 7 elements? This needs to be clearer in the methods section, and state that this is the primary endpoint, secondary tests are done for individual components of the score. I had thought the primary quantity of interest was the satisfaction at discharge time (after the intervention had occurred in the intervention group). Please clarify for all readers.

7. PLOS authors have the option to publish the peer review history of their article (what does this mean?). If published, this will include your full peer review and any attached files.

Reviewer #1: No

Reviewer #3: **Yes: **Dr. Esmail Khodadadi

Reviewer #4: No

---

## [Author Response · Author response to Decision Letter 1]

3 Apr 2024

Dear Reviewer

Thanks a lot for your mention and valuable comments. We tried to improve our manuscript.

Best regards

---

## [Decision Letter · Decision Letter 2]

22 Apr 2024

PONE-D-23-29600R2Effect of Implementing of the IDEAL Discharge Model on Satisfaction of Patient Referred to Trauma Emergency DepartmentPLOS ONE

Dear Dr. Ghafouri,

Thank you for submitting your manuscript to PLOS ONE. After careful consideration, we feel that it has merit but does not fully meet PLOS ONE’s publication criteria as it currently stands. Therefore, we invite you to submit a revised version of the manuscript that addresses the points raised during the review process.

We look forward to receiving your revised manuscript.

Kind regards,

Milad Khorasani, PhD

Academic Editor

PLOS ONE

Journal Requirements:

Reviewers' comments:

Reviewer's Responses to Questions

**Comments to the Author**

1. If the authors have adequately addressed your comments raised in a previous round of review and you feel that this manuscript is now acceptable for publication, you may indicate that here to bypass the “Comments to the Author” section, enter your conflict of interest statement in the “Confidential to Editor” section, and submit your "Accept" recommendation.

Reviewer #4: (No Response)

2. Is the manuscript technically sound, and do the data support the conclusions?

Reviewer #4: Yes

3. Has the statistical analysis been performed appropriately and rigorously? 

Reviewer #4: Yes

4. Have the authors made all data underlying the findings in their manuscript fully available?

Reviewer #4: Yes

5. Is the manuscript presented in an intelligible fashion and written in standard English?

Reviewer #4: (No Response)

6. Review Comments to the Author

Reviewer #4: Thank you for the revised manuscript. It is clearer and easier to understand now.

line 241 in track changed version: what is the unit for time in the emergency room? Assume minutes or hours, but please state clearly.

7. PLOS authors have the option to publish the peer review history of their article (what does this mean?). If published, this will include your full peer review and any attached files.

Reviewer #4: No

---

## [Author Response · Author response to Decision Letter 2]

24 Apr 2024

Dear Reviewer

Thanks a lot for your mention and valuable comments. We tried to improve our manuscript.

We added a unit for time in the emergency room (hours) in line 241.

Best regards.

---

## [Decision Letter · Decision Letter 3]

22 May 2024

Effect of Implementing of the IDEAL Discharge Model on Satisfaction of Patient Referred to Trauma Emergency Department

PONE-D-23-29600R3

Dear Dr. Ghafouri,

We’re pleased to inform you that your manuscript has been judged scientifically suitable for publication and will be formally accepted for publication once it meets all outstanding technical requirements.

Kind regards,

Milad Khorasani, PhD

Academic Editor

PLOS ONE

Additional Editor Comments (optional):

Reviewers' comments:

Reviewer's Responses to Questions

**Comments to the Author**

1. If the authors have adequately addressed your comments raised in a previous round of review and you feel that this manuscript is now acceptable for publication, you may indicate that here to bypass the “Comments to the Author” section, enter your conflict of interest statement in the “Confidential to Editor” section, and submit your "Accept" recommendation.

Reviewer #4: All comments have been addressed

2. Is the manuscript technically sound, and do the data support the conclusions?

Reviewer #4: (No Response)

3. Has the statistical analysis been performed appropriately and rigorously? 

Reviewer #4: (No Response)

4. Have the authors made all data underlying the findings in their manuscript fully available?

Reviewer #4: (No Response)

5. Is the manuscript presented in an intelligible fashion and written in standard English?

Reviewer #4: (No Response)

6. Review Comments to the Author

Reviewer #4: (No Response)

7. PLOS authors have the option to publish the peer review history of their article (what does this mean?). If published, this will include your full peer review and any attached files.

Reviewer #4: No

---

## [Editor Report · Acceptance letter]

4 Jun 2024

PONE-D-23-29600R3 

PLOS ONE

Dear Dr. Ghafouri, 

I'm pleased to inform you that your manuscript has been deemed suitable for publication in PLOS ONE. Congratulations! Your manuscript is now being handed over to our production team.

Kind regards, 

on behalf of

Dr. Milad Khorasani 

Academic Editor

PLOS ONE